# Detection and Monitoring of Small-Scale Diamond and Gold Mining Dredges Using Synthetic Aperture Radar on the Kadéï (Sangha) River, Central African Republic

**Marissa A. Alessi** [1,*] , **Peter G. Chirico** [1] , **Sindhuja Sunder** [2] and **Kelsey L. O'Pry** [3]

1    U.S. Geological Survey, Florence Bascom Geoscience Center, 12201 Sunrise Valley Drive, Reston, VA 20192, USA
2    Akima Systems Engineering, Contracted to U.S. Geological Survey, Florence Bascom Geoscience Center, 12201 Sunrise Valley Drive, Reston, VA 20192, USA
3    Natural Systems Analysts, Contracted to U.S. Geological Survey, Florence Bascom Geoscience Center, 12201 Sunrise Valley Drive, Reston, VA 20192, USA
*    Correspondence: malessi@usgs.gov

**Abstract:** Diamond and gold mining has been practiced by artisanal miners in the Central African Republic (CAR) for decades. The recent introduction of riverine dredges indicates a transition from artisanal/manual digging and sorting techniques to small-scale mining methods. This study implements a remote sensing analysis of Synthetic Aperture Radar (SAR) data to map gold and diamond dredges operating on the Kadéï (Sangha) river in the CAR. Riverine vessels are identified in Sentinel-1 SAR data between 2015 and 2019, and their activity levels are mapped over time. The number of active dredges identified on the river increased over the five years studied, with the largest increase occurring between 2016 and 2017. Detailing a method for mapping and monitoring riverine diamond and gold dredge mining is an important step in keeping up with evolving technologies and new areas of mineral exploitation and in helping address concerns over resource governance in remote and conflict-prone terrain. The use of SAR technology, with its weather-independence, broad coverage, and available wavelength combinations, allows for higher temporal resolution and improved vessel detection in the monitoring of small-scale mining (SSM) dredges.

**Keywords:** Synthetic Aperture Radar (SAR); dredges; artisanal and small-scale mining (ASM); radar detection; remote sensing; gold mining

## 1. Introduction

### 1.1. Mining History

Located just north of the equator (Figure 1), the Central African Republic (CAR) has been known and exploited for its rich mineral and natural resources for over 100 years. Primary resource interests in the CAR have, in the past, included rubber, coffee, cotton, diamonds, and gold, all of which remain, to varying extents, contributors to the country's economy today [1]. Since the CAR's independence from France in 1960, successive governments have depended heavily on the mining sector, and on diamond and gold exports in particular, to bring in revenue. CAR heads of state have pursued both domestic capacity-building and foreign investment and technology transfer in gold and diamond extraction over the years. In the late 1960s and 1970s, the government solicited Israeli investment in the CAR diamond sector and both German and private-sector diamond-processing know-how [2]. In the 1970s and 1980s, CAR citizens were encouraged to set up individual, artisanal diamond and gold mining operations [3]. Export taxes were lowered and the Bureau d'évaluation et de Contrôle de Diamant et d'Or (BECDOR) was created to oversee a diamond and gold tracing system [4]. Between 1993 and 2013, a system of mining licenses, mining cooperatives (société miniers), middlemen (collecteurs), and buying



houses (bureaux d'achat), all of which report to BECDOR, was instituted [4]. The CAR is also a participant in the Kimberley Process (KP) and has been accepted as a member of the Extractive Industries Transparency Initiative (EITI) [4–6]. The CAR Mining Code was first established in 1961 and has been revised several times since then, including in 1979, 1986, 2004, and 2009, with another revision underway as of 2021 [6,7].

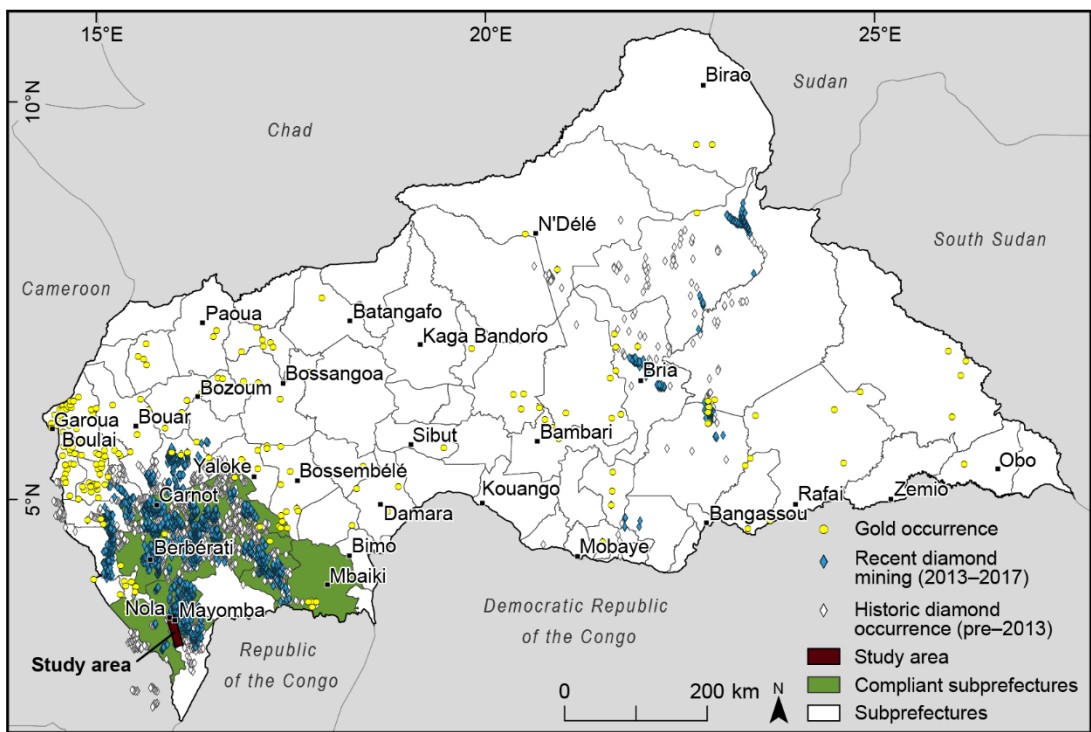

**Figure 1.** Location map of the study area within the southwest region of the CAR, showing the eight Kimberley Process-compliant subprefectures as of 2019. Diamond occurrence and mining data from [5]. Gold occurrence data from [8].

The CAR's natural resources sector has been impacted by internal conflict, changing governance regimes and priorities, international sanctions, and smuggling [3,9–11]. Control of diamond and gold mining and mine sites has been contested throughout the CAR's ongoing, decade-long civil war, and the KP suspended the CAR's ability to legally export diamonds in 2013 to avoid tacitly endorsing the sale of blood or conflict diamonds [5,10]. Because the production of gold was not similarly restricted, many artisanal diamond miners switched their focus, increasing the prevalence of gold mining [12]. As conflict dissipated in the CAR's west and government control was reestablished, the five subprefectures of Berbérati, Nola, Carnot, Gadzi, and Boda were reopened for diamond production between 2015 and 2016 [13]. Three more, Boganangone, Mbaïki, and Boganda, were approved in 2019 for a total of eight KP-compliant subprefectures [14].

In 2020, gold and diamonds comprised 38% of the CAR's exports, accounting for 34.7 million USD and 14.7 million USD, respectively [1]. As of 2019, the CAR's artisanal and small-scale mining (ASM) sector directly employed about 5% of the country's population and added between 55 million USD and 80 million USD to the rural economy yearly [6,15]. Figure 1 depicts the extent of gold and diamond mining and mineral occurrences in the CAR between 2007 and 2017. Multiple accounts over the years have indicated that significant quantities of gold and diamonds leave the country unreported [4,5,9,16–19]. The smuggling of gold and diamonds appears to result from a combination of factors, including the informality of the mining sector, the scattered nature of the deposits, mine site distance from law-enforcement spheres of control, and porous national borders [5,17,19].

### 1.2. Mining Practices

All the CAR's diamond deposits are alluvial in nature, occurring within the Carnot Sandstone and the Mouka-Ouadda Sandstone formations [5]. Diamonds are found in channel, alluvial flat, and terrace deposits and are extracted manually through artisanal or semi-mechanized methods using hand tools—picks, shovels, and sieves—coupled with the use of water pumps to drain the excavated pits [5,17,20,21]. The CAR's gold resources are found in both alluvial and hard-rock deposits and are recovered by employing mechanized dredges along riverbanks or through the digging of vertical shafts [12]. Between 1995 and 1998, the Projet d'appui au secteur artisanal du diamant en République Centrafricaine (PASAD project), undertaken by the Bureau de Recherches Géologiques et Minières (BRGM: the French Geological Survey), conducted experiments with small, 10-cm (4 in) diameter suction dredges on the Mambéré river south of Nola. The results showed dredge mining to be very efficient, with a 90% recovery rate for heavy minerals and diamonds [22,23]. However, no mining companies are known to have employed dredging in the CAR until the introduction of bucket-line river dredges between 2012 and 2015 [24].

The two types of dredges most used for alluvial channel mining are the bucket-line dredge and the suction dredge (Figure 2) [25]. The bucket-line dredge (Figure 3) uses a chain of large, fortified buckets that travel in a loop along a conveyor belt, digging into the river bottom. The buckets carry this sediment onboard, where it is passed through classifying screens that sort and separate gravel, clay, grass, and branches from finer material. This material is then run through pulsating and fixed sluice boxes to extract gemstones, fine gold particles, and gold nuggets, with the unwanted gravel and sediment, known as tailings, redirected away from the work site into spoil piles [26]. Suction dredges utilize a boom with a rotating claw that is lowered into the river to break up rocks and consolidated materials and churn up channel sediments. The sediments are collected by a suction tube and sorted by grain size using carpets and sluice boxes that act as sieves [27]. Tailings are deposited along the stream bed. Small-scale gold mining dredges frequently employ mercury for fine gold amalgamation and recovery; the mercury is either added to the sluice box or applied to the concentrate after sluicing. In either case, the use of mercury is a human and environmental health concern [12]. In the CAR, dredge mining primarily targets gold deposits, though diamonds can be recovered as bycatch [9]. Both bucket-line and suction dredges have been observed on rivers in the western CAR [15].

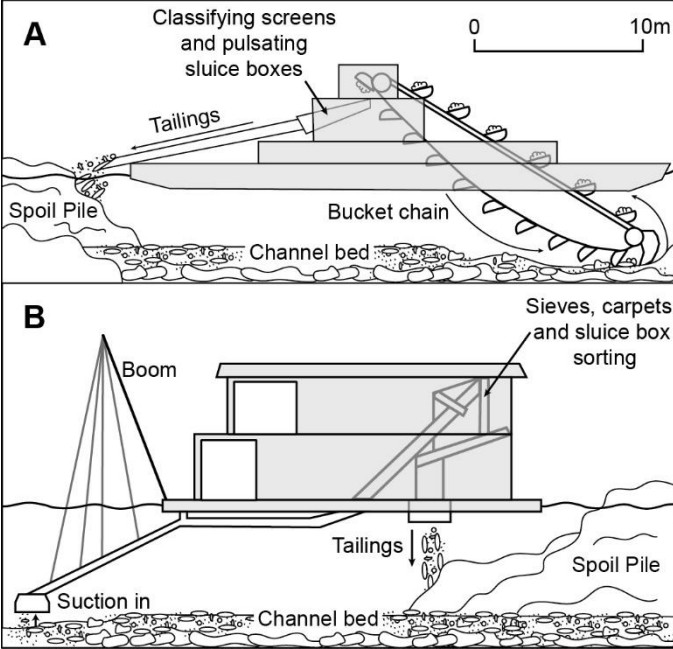

**Figure 2.** Structure of a bucket-line dredge (**A**) and a suction dredge (**B**).

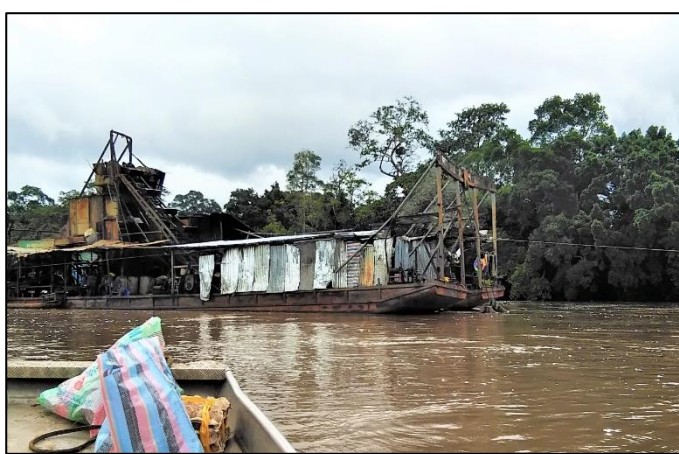

**Figure 3.** A bucket-line mining dredge on the Kadéï (Sangha) river in 2019. Photo credit: Ministère des Mines et de la Géologie de la République Centrafricaine.

The introduction of dredge mining in the CAR denotes a significant transition from artisanal mining to small-scale mining (SSM) techniques. Dredges represent a new mining frontier, opening previously unexploited areas to mining. To fully understand the nature of diamond and gold extraction in the CAR, it is necessary to develop methods to map and monitor ASM, including dredge movements and dredge mining. Identifying dredges and hotspots of dredge activity is valuable in deriving gold and diamond production estimates and assessing when and where the environmental impacts of ASM may occur. Advanced methods to track dredge mining can also contribute to larger efforts in mapping and monitoring ASM activity globally.

Many studies have made use of satellite imagery to map and monitor ASM, whether of gold, diamonds, or other commodities [5,28–32]. Remote sensing methods and related geospatial modelling techniques are ideally suited to mapping mining and production in remote and conflict-prone regions. This paper proposes and tests the use of Sentinel-1 Synthetic Aperture Radar (SAR) data to identify and monitor the activity of small-scale gold and diamond mining dredges operating in riverine environments. We apply geospatial modelling and remote sensing techniques to assess dredge activity within a 30-km reach of the Kadéï (Sangha) river between 2015 and 2019. The method proposed here has potential for application worldwide, wherever dredge mining is found.

### 1.3. SAR Background

SAR is an active remote sensing technology that works by emitting an electromagnetic signal (between 2 and 100 cm wavelengths) and measuring the strength of the energy returned from the Earth's surface, known as backscatter. A SAR satellite can have either ascending—moving from south to north—or descending—moving from north to south—flight tracks and, thus, may observe phenomena from opposing viewing directions. The angle of the line of sight, known as the slant range, varies across the ground extent of the SAR scene, known as the swath width (Figure 4) [33]. As an active sensor that uses longer wavelengths than optical sensors, SAR has the advantage of being daylight- and weather-independent [34]. This makes it particularly useful in areas with large gaps in optical data due to persistent cloud coverage. Longer wavelengths also allow SAR to penetrate farther through surface features (e.g., through topsoil or through a forest canopy to understory structures) than optical sensors [33]. Backscatter interactions with features on the ground are affected by signal wavelengths, referred to as bands, and the orientation of the sending and receiving plane, referred to as polarization [35]. The polarization of a SAR wavelength is represented by a trigonometric combination of the variables H (send or receive horizontal) and V (send or receive vertical).

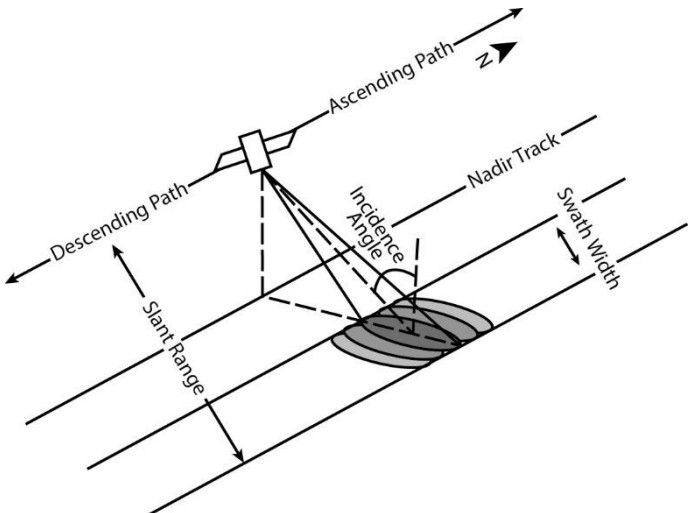

**Figure 4.** Typical SAR imaging geometry.

Surface parameters strongly influence backscatter strength and appearance in SAR data. Flat surfaces in SAR data typically produce weak-to-no backscatter, called specular reflection, as the SAR signal is reflected away from the receiving satellite (Figure 5). Rough surfaces increase backscatter by redirecting the signal in many directions, known as diffuse scattering [35]. Man-made objects with right-angles, like buildings or boats, can produce very strong backscatter signals, known as double-bounce or corner reflections, as the SAR pulse is reflected directly back to the sensor [36]. In addition, backscatter strength is influenced by the material with which the SAR signal interacts; for example, a greater amount of energy is returned by materials with a high dielectric constant, like metal, than by, say, concrete [35]. Backscatter strength is also affected by the angle between the SAR signal and the earth's surface, referred to as the incidence angle or angle of interaction [35]. In combination, imaging and surface properties create measurable differences in backscatter that lead to variable degrees of energy being returned to the sensor. These differences in energy, represented by pixel values, can be used to set thresholds for classifying land cover or identifying objects of interest that may not be discernible in other types of imagery.

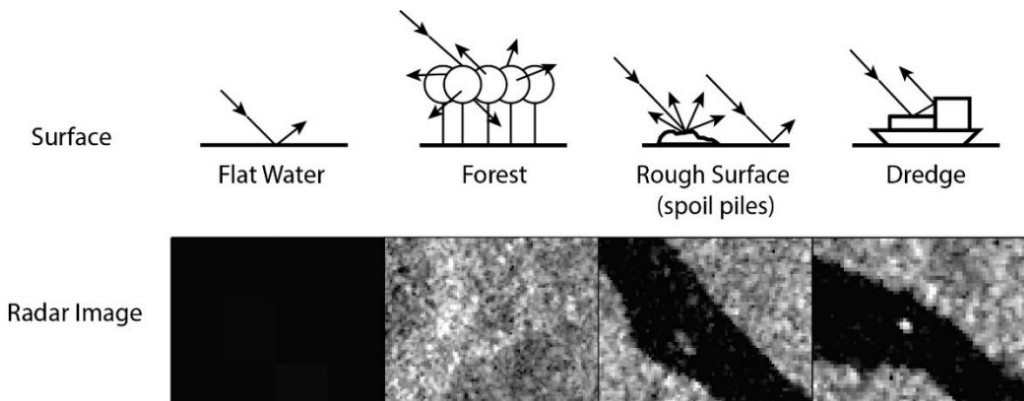

**Figure 5.** SAR interaction with different surface features and their appearance in the data. VV channel SAR data from Copernicus Sentinel data [2019]. Retrieved from ASF DAAC [14 June 2021], processed by ESA.

A number of commercial and government SAR missions are in progress or are planned for the near future [37–39]. Their use of different SAR wavelengths and polarizations means they represent new capabilities for higher spatial and temporal resolution and broader radar data coverage. A brief discussion of three constellations and missions is included

here as a forward-looking view of how SAR facilities and data availability can further improve the ability to map and track mining dredges operating in riverine environments across the globe.

Capella Space is a U.S.-based commercial data provider with a constellation of seven SAR satellites capable of collecting single polarization (HH or VV), very-high-resolution X-band (3-cm wavelength) SAR data at ground resolutions of 0.5–1.5 m (mode dependent) [37]. Their satellites orbit in both ascending and descending passes and provide left- and right-looking SAR data. Similarly, ICEYE is a commercial provider, based in Finland, with a constellation of over 18 SAR satellites. Their sensors are X-band (3 cm) and VV polarized, with image products having 1 m, 3 m, and 15 m ground resolution, depending on the collection mode [38]. Figure 6 compares the 3 m image resolution of ICEYE to Sentinel-1 10 m image resolution. Details like the shape, size, and wake of the vessel, where wake indicates direction of travel, are sharper in the ICEYE data. As with Capella Space satellites, ICEYE satellites have ascending and descending orbits that provide left- and right-looking data. Finally, NISAR is a joint NASA and Indian Space Research Organization SAR mission with a planned launch date in 2024. The NISAR mission will collect dual-polarization (HH/HV or VV/VH) and dual band (24 cm L-band data and 10 cm S-band data) SAR products with a ground resolution of 3–10 m, mode dependent, with left-looking viewing geometry only [39]. The data will be publicly released at no charge.

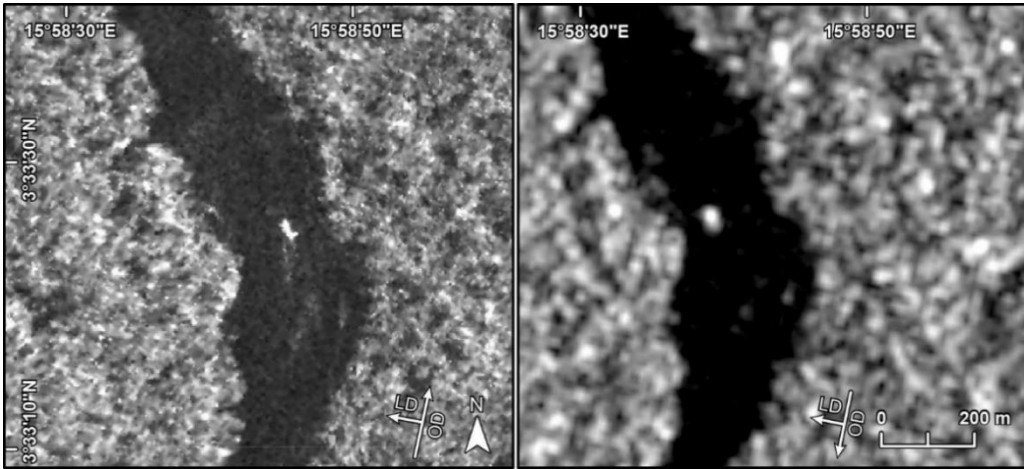

**Figure 6.** Comparison of dredge appearance in X-band, 3 m resolution, VV channel ICEYE data collected on 27 November 2020 (**left**) and C-band, 10 m resolution, VV channel Sentinel-1 data collected on the same day and at the same location (**right**). Data: (**left**) ICEYE Stripmap Mode [2020], (**right**) Copernicus Sentinel data [2020]. Retrieved from ASF DAAC [29 April 2022], processed by ESA. Arrows in the bottom right corners indicate the look direction (LD) and the orbit direction (OD) of the SAR scene.

*1.4. Ship Detection*

Due to differences in how SAR interacts with flat surfaces and man-made objects, it is commonly implemented in ocean ship detection studies [34,40–47]. Methods take advantage of SAR's weather-independence, wide swath width, and high temporal resolution to identify the large, metallic, and angular structures of ocean vessels, which produce strong backscatter in contrast with the weak backscatter returned by flat ocean surfaces [34]. Ship detection studies use SAR satellites with X-, C-, and L-band sensors, with the most common being C-band due to the wide coverage and easy availability of data from the Sentinel-1 satellite [42,44,45,48–51]. Polarization varies from study to study based on data availability and the imaging properties of the scene. Greidanus et al. [43] indicates that in the case of dual polarized (HH and HV or VH and VV) satellites like Sentinel-1, a combination of the co- (VV/HH) and cross- (HV/VH) polarized channels yields the optimal conditions for ship detection; whereas either cross-polarized channel in combination with a VV channel

will work, a cross-polarized channel and an HH channel provides higher ship-sea contrast. Leng et al. [51] writes that cross-polarized channels have higher ship-sea contrast and are better suited at identifying ships when the SAR data has an incidence angle below 45°, while co-polarized channels have less clutter and are therefore more suitable for ship identification at incidence angles greater than 45°. Leng et al. [51] also notes that, when only co-polarized data is available, HH is the better option.

Many ship detection methods are fundamentally based on differentiating strong signals (from corner reflectors) from weak signals (from flat water) [50]. Most ship detection approaches use adaptive thresholds and automated classification detection techniques to compare the intensity and statistical properties of individual pixels with that of their local neighborhoods to identify pixels of unusual strength [34,43,44,49,51,52]. This approach should be transferable to riverine environments, especially as river surfaces are generally much calmer and flatter than ocean surfaces. However, riverine vessels tend to be smaller than ocean vessels, with a greater chance of overall backscatter interference from land and other manmade structures due to the reduced water surface areas in play. In addition, riverine dredges are much more elusive than ocean vessels; they do not necessarily report their locations and generally operate independently and unpredictably.

With the exception of Gruel and Latrubesse [53], ocean ship detection methods utilizing SAR have yet to be widely applied to riverine environments. Gruel and Latrubesse [53] successfully implements a backscatter thresholding technique and time-series analysis to pinpoint riverine vessel locations on the Irrawady river using C-band SAR data from the VV channel of the Sentinel-1 satellite, thus proving that riverine vessels can be observed in Sentinel-1 data and that the backscatter produced by smaller vessels is distinguishable from river surfaces. However, they [53] acknowledge the challenge of distinguishing vessel type and tailor their study to the specific habits of sand dredges to reduce false identification; their observations occur only in the central parts of the river and only at times when sand dredges are known to be active. Gold and diamond mining dredges, on the other hand, run differently, hugging riverbanks and operating at ever-changing times and places.

## 2. Materials and Methods

### 2.1. Study Area

The recent presence of dredges on rivers in southwestern CAR suggests a new method for recovering gold and diamonds from river channels and riverbanks. Reports indicate that these dredges are largely owned and operated by small-scale Chinese companies [9,15,54,55]. To address the need for consistent monitoring of this new type of mining activity, this study analyzes dredge mining between 2015 and 2019 along a 30 km reach of the Kadéï river in the KP-compliant subprefecture of Nola (Figure 7). A methodology of data acquisition, manual digitization and validation, and density surface analysis is implemented (Figure 8).

### 2.2. High-Resolution Imagery

Though mining dredges are clearly distinguishable in traditional high-resolution optical satellite imagery of the study area, consistent tracking is hindered by the frequent presence of cloud cover in the images, reducing the temporal resolution of any subsequent analysis. Consequently, this study uses high-resolution optical imagery only to verify the presence of mining dredges on the river, to validate results obtained from SAR analysis (Figure 9), and to increase confidence in the visual cues used to search the SAR scenes (Figure 10).

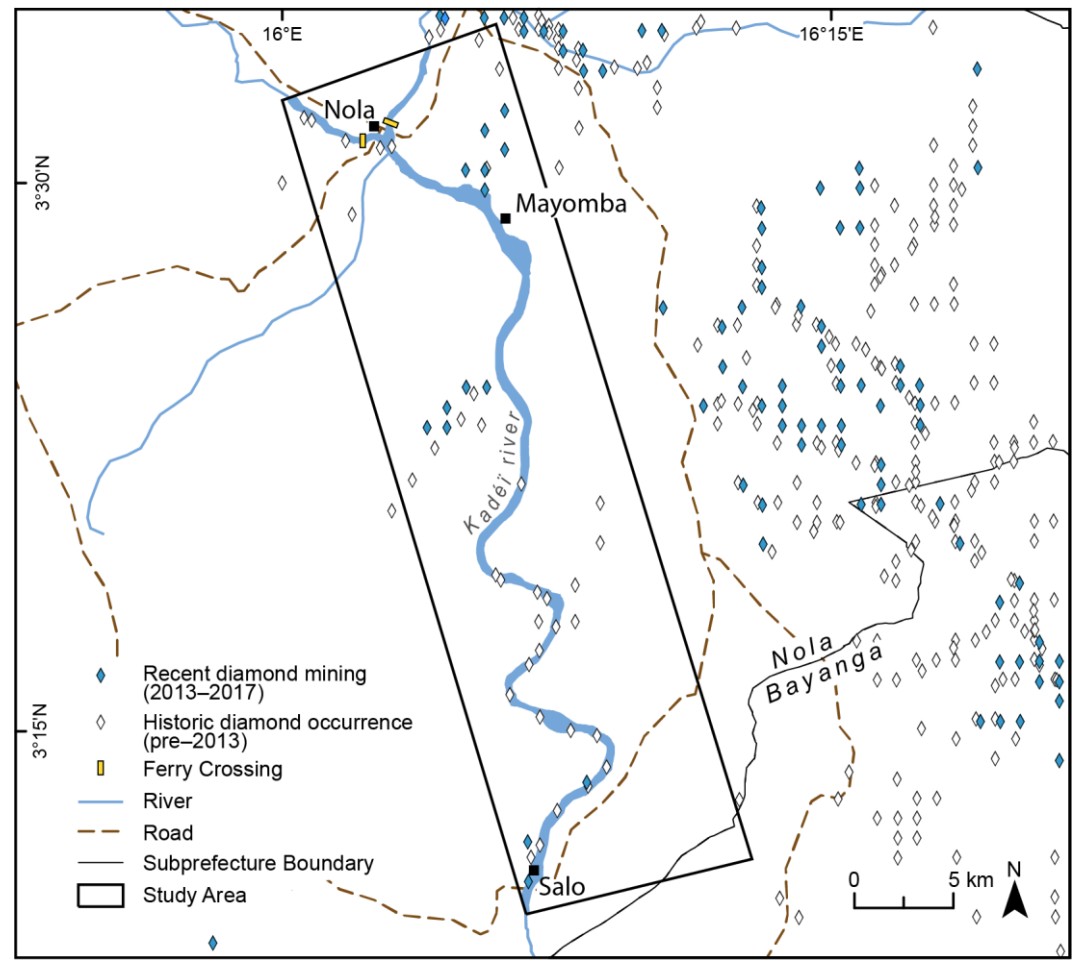

**Figure 7.** Close up of the study area and proximate diamond mining in the subprefecture of Nola. Diamond occurrence and mining data from [5].

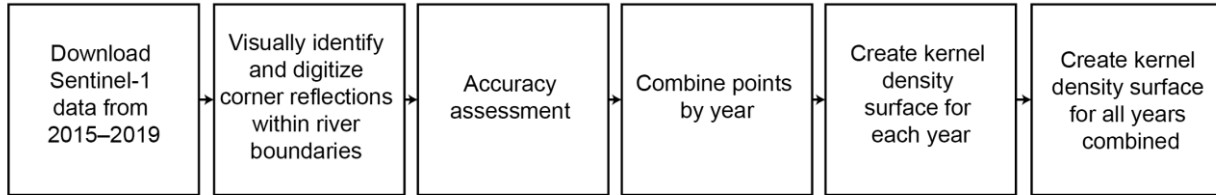

**Figure 8.** Methodology flow chart.

World View-2 (WV02) optical satellite imagery distributed by Maxar (previously DigitalGlobe) provides a high-resolution panchromatic band and eight multispectral bands at 0.46 m resolution. Figure 9, a high-resolution WV02 pan-sharpened natural-color image taken on 15 April 2020, shows a bucket-line-type gold and diamond dredge with a blue roof on the Kadéï just south of Nola. The size and configuration identify it as a bucket-line dredge with two sets of arms and pontoons measuring 32 m × 2.2 m × 1.15 m. These dredges have an operating capacity of 100 tons/hour and a dredging depth of 15 m [56]. A similar dredge appears in the same location in WV02 imagery dated 1 November 2019 and 29 November 2019 in Figure 10. Other dredges of a smaller size, 25 m long, also appear in these high-resolution images; these are older dredges similar to that seen in Figure 3. Though shorter, they have two sets of arms and a comparable structure to that of the blue dredge visible in Figure 9. Figure 9 also depicts smaller, 16 m long vessels, known as tender boats, that service dredges and assist in fuel and cargo transport.

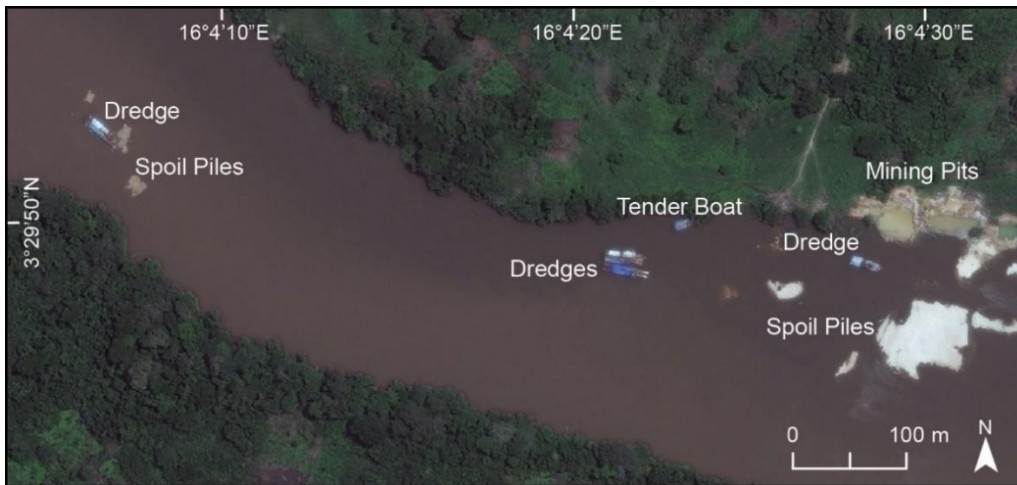

**Figure 9.** High-resolution imagery showing mining activity, including dredges, on the Kadéï river, taken on 15 April 2020. World View-2 (WV02) satellite imagery from Maxar [2020]. Retrieved from EVWHS [5 July 2021].

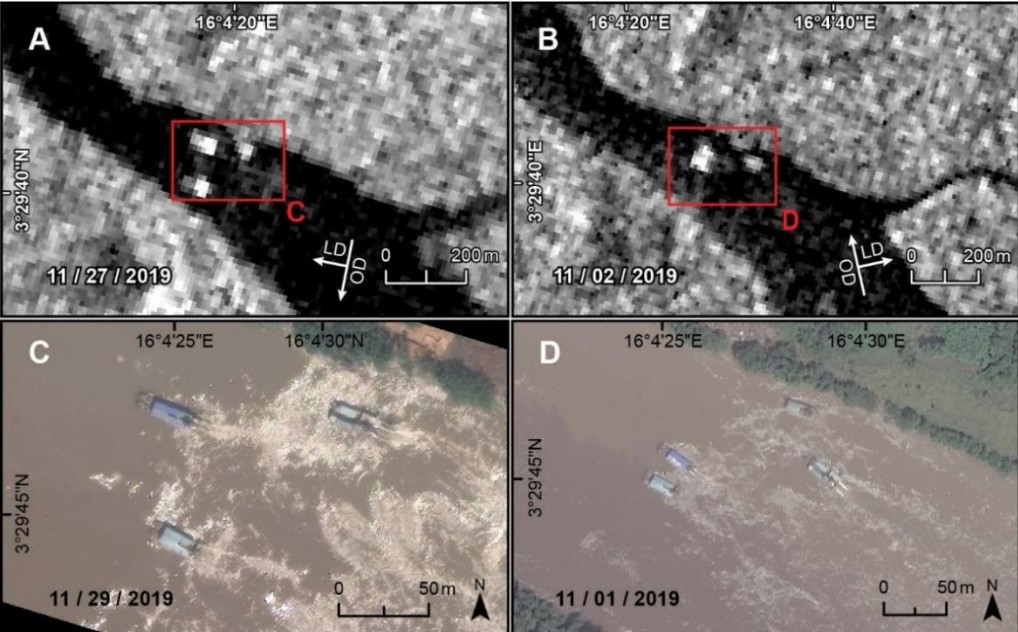

**Figure 10.** Example of riverine vessels in Sentinel-1 VV channel SAR and high-resolution imagery taken within two days of each other. (**A**,**B**) SAR data from Copernicus Sentinel data [2019]. Retrieved from ASF DAAC [14 June 2021], processed by ESA. (**C**,**D**) World View-2 (WV02) satellite imagery from Maxar [2019]. Retrieved from EVWHS [14 July 2021]. Arrows in the bottom right corners indicate the look direction (LD) and the orbit direction (OD) of the SAR scene.

## 2.3. Sentinel-1 SAR

Sentinel-1 SAR data distributed by the European Space Agency (https://scihub.copernicus.eu/dhus/#/home) was collected from the Alaska Satellite Facility Distributed Active Archive Center (ASF DAAC) using the ASF Search Vertex (https://search.asf.alaska.edu/#/ (accessed on 5 July 2021)). The Sentinel-1 SAR instruments include twin satellites that orbit 180° apart, observing the entire earth every six days [48]. Sentinel-1 SAR uses wavelengths in the C-band (5.5 cm) that have low-to-moderate surface feature/foliage penetration capabilities. This study uses Sentinel-1 SAR Ground Range Detected Interferometric Wide Swath products, which have a 250 km swath at a 5 m × 20 m spatial resolution (slant-range and along-track, respectively) and are dual-polarized with VV and

VH channels. Incidence angles range from 15° to 45° and include both ascending and descending flight tracks. Every scene available on the ASF Search Vertex covering the study area between 10 January 2015 and 27 December 2019 was acquired and analyzed (Table 1). This amounts to 350 dates and 387 scenes, with a scene spacing of 1–6 days between observations.

**Table 1.** Table of collected Sentinel-1 SAR scenes broken down by year, flight direction, and path and frame numbers.

| Year | Days Observed | Total Scenes | Number of Ascending Scenes | Number of Ascending Paths/Frames | Number of Descending Scenes | Number of Descending Paths/Frames |
|---|---|---|---|---|---|---|
| 2015 | 26 | 32 | 19 | 88/1193 88/1187 88/1188 | 13 | 7/580 7/581 7/583 7/578 |
| 2016 | 61 | 89 | 56 | 88/3 88/9 88/1187 88/1193 | 33 | 7/580 7/582 7/583 7/578 |
| 2017 | 86 | 89 | 32 | 88/3 88/5 88/9 88/1189 | 57 | 7/580 7/581 7/582 |
| 2018 | 88 | 88 | 30 | 88/1189 | 58 | 7/581 7/582 |
| 2019 | 89 | 89 | 30 | 88/1189 | 59 | 7/581 7/582 |
| Total | 350 | 387 | 167 | | 220 | |

### 2.4. Manual Detection

Riverine vessels on the Kadéï produce a significant SAR backscatter signature compared to surrounding surface water and are clearly visible operating within the study area (as seen in Figures 5, 6, 10 and 11). Their size, structure, and material return strong and prominent corner reflections in comparison to the low backscatter returns caused by the specular reflection of the surface water. The backscatter values detected in the VV channel were found to be much stronger and larger than those in the VH channel. Co-polarization usually presents stronger backscatter due to volume scattering, making it easier to identify vessel signals when visually scanning a scene.

In addition to qualitative differences in backscatter returns, quantitative variations in SAR data intensity values may also help identify a potential vessel. The intensity values of potential vessels were found to be in the hundreds of thousands, with the highest values in the center of the cluster and the edges staying above 100,000. The intensity of water pixels stayed below 10,000. The intensity of small islands and spoil piles was largely in the 10,000 s, though sometimes above 100,000 (but never above 200,000).

Only the areas within the river and along the riverbanks were considered in the visual search for vessels, which, once manually identified, were digitized as point features (Figure 11). The Kadéï sees virtually no boat traffic apart from dredge mining, a ferry barge in Nola, and limited tender boat activity in support of the dredges. Minimal non-dredge occurrences on the river reduce the chance of an identified vessel being something other than a mining dredge. As a further check, WV02 imagery of dredges operating on the river was compared with SAR data collected in the same time-frame to increase confidence in the identification cues used during visual interpretation. Figure 10 is an example of such a comparison; dredges appear in high-resolution imagery and in SAR data in the same location within no more than 2 days of each other, confirming the accuracy

of the identification methodology and this study's understanding of SAR signals. Where available, WV02 imagery was also used as a reference to help ensure that strong but ambiguous backscatter returns were not caused by an object other than a dredge, such as a spoil pile or dock along the riverbank.

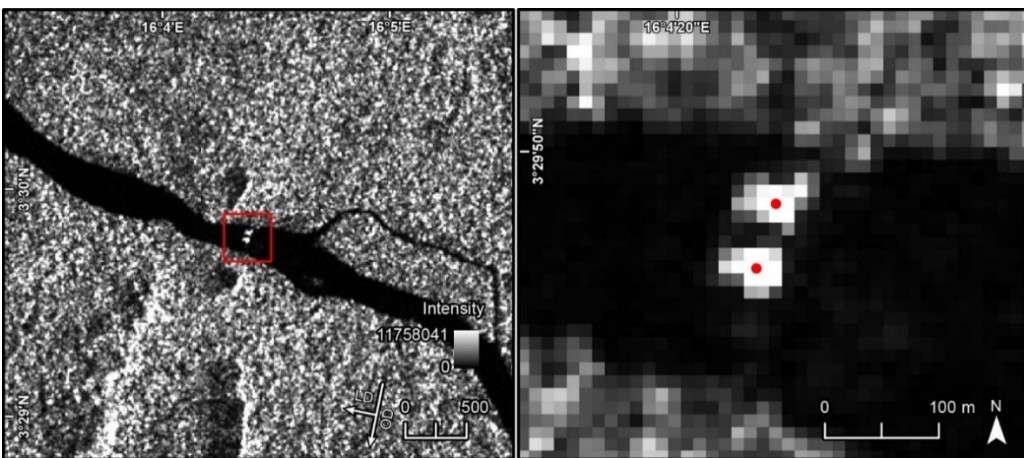

**Figure 11.** Example of riverine vessels in Sentinel-1 VV channel data being digitized as point features through visual interpretation. The red square in the left figure depicts the area shown in the right figure and the red dots represent the digitized features. SAR data, 22 October 2019, from Copernicus Sentinel data [2019]. Retrieved from ASF DAAC [14 June 2021], processed by ESA. Arrows in the bottom right corner indicate the look direction (LD) and the orbit direction (OD) of the SAR scene.

### 2.5. Density Analysis

Once all 387 SAR scenes were surveyed and a point dataset for each year was digitized, a kernel density surface calculating the number of vessel appearances per square kilometer was created for each year. Any vessel that appears in the same location in all the SAR scenes for that year was labelled as idle. Any vessel that was in the path of the known ferry-crossing and appeared to travel back and forth across the river consistently was labelled a ferry. The remaining points were considered active dredges. Neither the ferry nor the idle category was used in the final density surface-creation process. As a final step, a combined density surface of all possible dredge activity between 2015 and 2019 was created from the entirety of the digitized points (excluding the ferry and idle categories).

### 2.6. Accuracy Assessment

Two accuracy assessments were performed to measure the success of this study's methodology of manual detection of ASM dredges in SAR data. The first assessment utilized the 0.46 m resolution WV02 imagery, in which dredges can be confidently distinguished from other types of vessels on the river, allowing for an absolute accuracy assessment. However, there were only two incidences wherein WV02 imagery and SAR scenes shared a date. Limited data availability, patchy coverage, and cloud interference in the WV02 imagery meant that a second accuracy assessment was necessary. This assessment used Planet Labs imagery, which has a lower spatial resolution of 3 m, but more consistent coverage [57]. The 3 m resolution does not provide enough detail to distinguish vessel type but can be used to confirm the presence of a vessel on the river. With wider temporal coverage, Planet Labs imagery corresponded with SAR scenes on 13 separate dates.

For both assessments, any vessel on the river or by either riverbank was digitized as a point feature and marked with the date the imagery was taken. Digitized points from the WV02 imagery were also attributed with vessel type. These accuracy points were then compared with the points identified in the SAR scene that shared their date of acquisition. Each point was labelled True Positive (TP) if the point in the optical imagery and the point in the SAR scene were a match; in the cases of the WV02 imagery, both points had to be

identified as dredges, while in the case of the Planet Labs imagery, a dredge identified from the SAR data had to be verified by the presence of any vessel in the optical imagery. A False Positive (FP) designation was granted if the point in the SAR scene identified a vessel of some type but was not matched by a point in the optical imagery. A False Negative (FN) designation was granted if a point identifying a dredge or vessel was digitized from the optical imagery but not in a corresponding SAR scene. The overall accuracy was calculated by dividing the number of points marked TP by the total number of accuracy points.

## 3. Results

In total, 1747 riverine vessels were identified in the 387 SAR scenes analyzed (Table 2). Of these points, 999 were determined to be active dredges. Dredge identifications in 2015 and 2016 numbered 28 and 35, respectively. There was a significant increase in identifications after 2016, with 285 identified dredges in 2017, 265 identified dredges in 2018, and 386 identified dredges in 2019. Table 2 and Figure 12 depict the yearly average and maximum number of active dredges identified in a SAR scene. An upward trend in dredge numbers is apparent. Point feature locations of the detected dredges are available in Alessi [58].

**Table 2.** Summary table of point results from the visual interpretation of collected SAR scenes.

| Year | Number of SAR Scenes | Total Vessel Identifications | Total Dredge Identifications | Average Number of Active Dredges | Maximum Number of Active Dredges |
|---|---|---|---|---|---|
| 2015 | 26 | 68 | 28 | 1.1 | 2 |
| 2016 | 61 | 169 | 35 | 1.3 | 3 |
| 2017 | 86 | 486 | 285 | 3.4 | 7 |
| 2018 | 88 | 466 | 265 | 3 | 5 |
| 2019 | 89 | 558 | 386 | 4.3 | 9 |

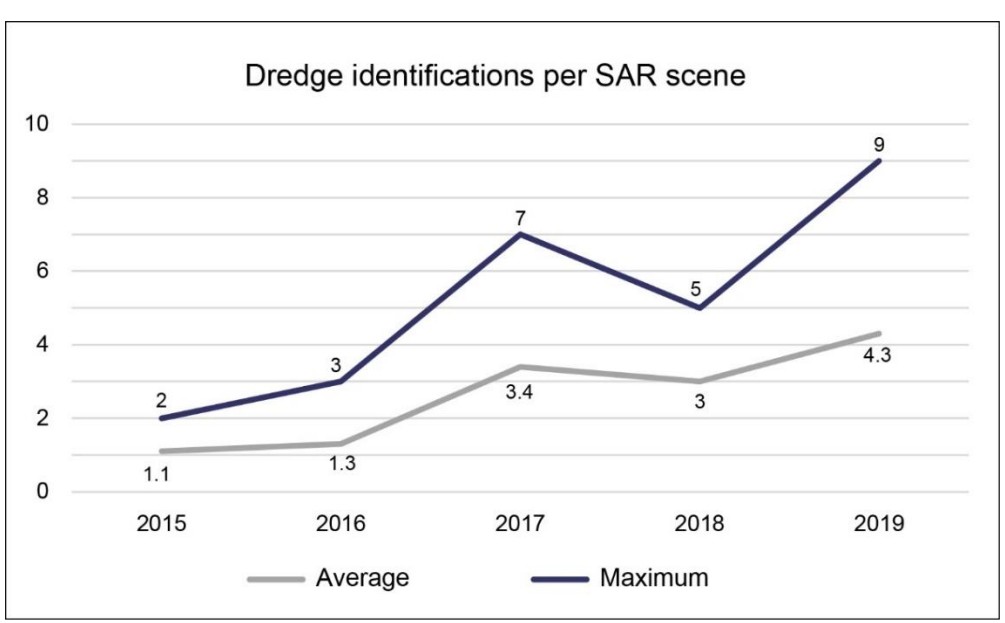

**Figure 12.** Average number of active dredges and maximum number of active dredges identified in a SAR scene for each year studied.

Of the two accuracy assessments conducted to evaluate the reliability of this study's visual detection of dredges in SAR data, the first assessment, using 0.46 m resolution WV02 imagery, resulted in six points (Table 3). Five of these points were accurately identified in the SAR data. The one misidentification indicates an 83% overall accuracy rate. The second assessment, with 3 m resolution Planet Labs imagery, allowed for the observation and

evaluation of 13 dates and 75 points. This assessment resulted in 54 TP attributions, 16 FP attributions, and five FN attributions for an overall accuracy of 72% (Table 3).

**Table 3.** Summary table of the accuracy assessment points and their overall accuracy.

|  | Accuracy Points | True Positive | False Positive | False Negative | Overall Accuracy |
|---|---|---|---|---|---|
| WV02 | 6 | 5 | 1 | 0 | 83% |
| Planet Labs | 75 | 54 | 16 | 5 | 72% |

Five density surfaces, one for each year in the study, were created from the active dredge points identified in the SAR scenes (Figure 13). An aggregate density surface for all study years combined was also created (Figure 14). The detection and density analyses show an increase in riverine vessel activity on the Kadéï over time. It is clear from the density analysis that dredging activity has targeted the southern and central portions of the study area throughout the entire period studied. Over the years, however, the concentration of riverine dredging activity moves north; in 2019, significant dredging activity was concentrated in the northern portion of the study area between Mayomba and Nola.

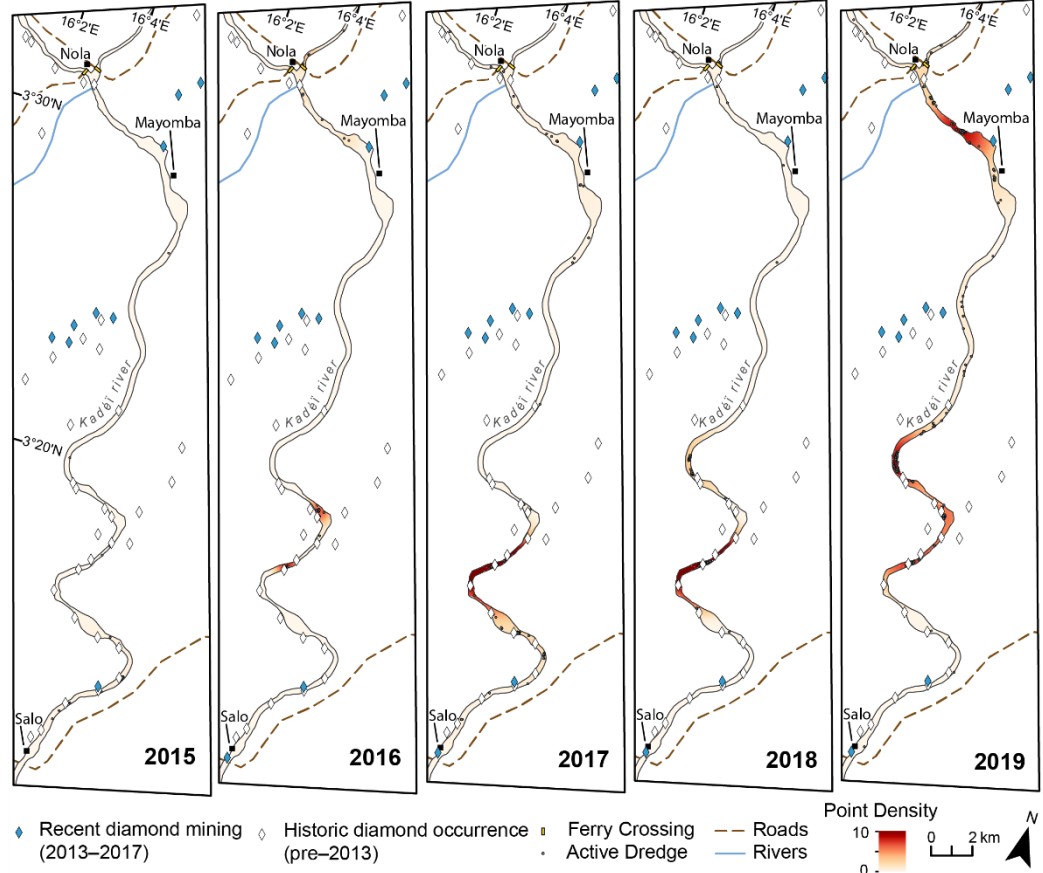

**Figure 13.** Kernel density surface of identified riverine vessels (point density) showing the number of vessel appearances per square kilometer for each year of the study. Diamond data from [5].

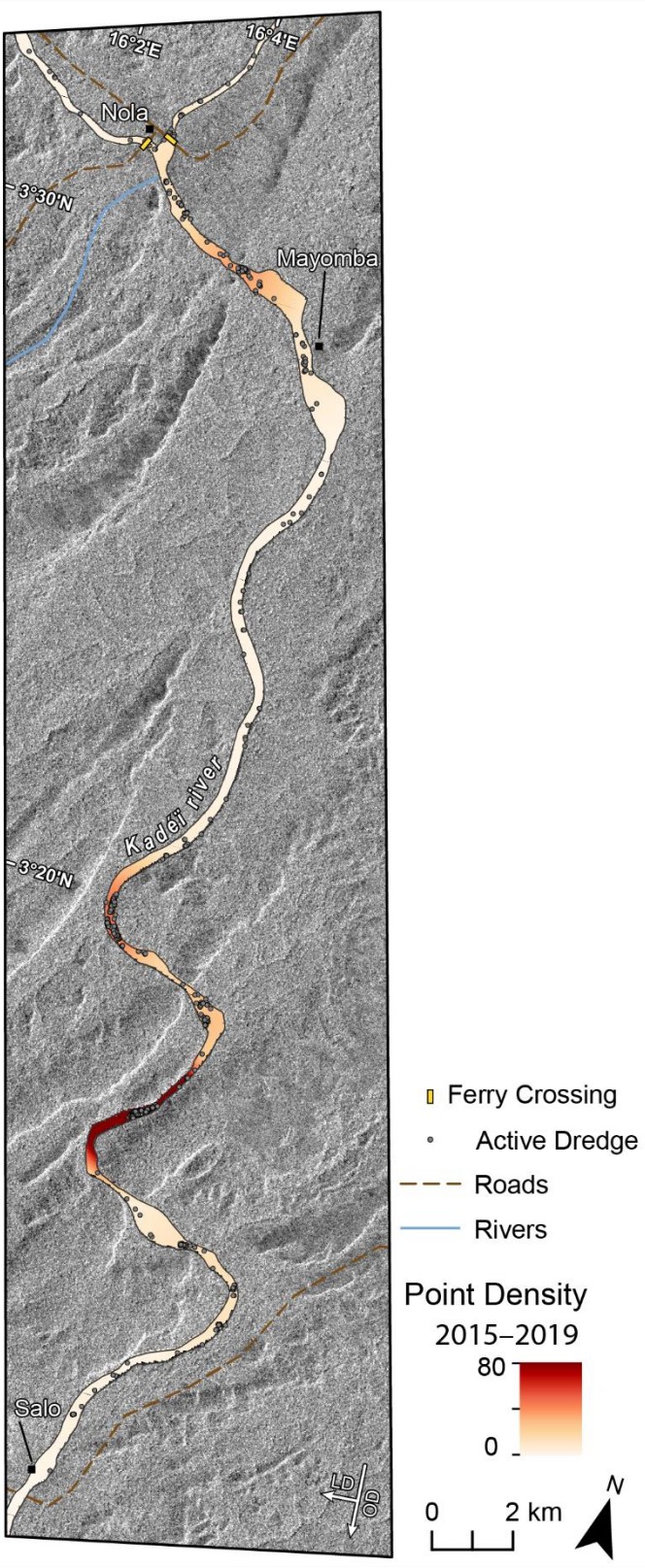

**Figure 14.** Kernel density surface of identified riverine vessels (point density) showing the number of vessel appearances per square kilometer for all years in the study: 2015–2019. VV channel SAR data from Copernicus Sentinel data [2019]. Retrieved from ASF DAAC [14 June 2021], processed by ESA. Arrows in the bottom right corner indicate the look direction (LD) and the orbit direction (OD) of the SAR scene.

## 4. Discussion

*4.1. SAR Advantages for Dredge Identification*

ASM activities are elusive in nature and are rarely or inconsistently reported formally. There is a particular lack of data on riverine dredge mining due to the frequent movement of mining vessels and remoteness of mining operations. Cloud cover makes it difficult to rely on optical imagery to identify and consistently monitor dredge activity. The wavelength capabilities of SAR allow for high temporal resolution and, thus, the ability to monitor ASM consistently over time without interference from frequent cloud cover. The size, material, and structure of active mining dredges create strong backscatter returns that contrast with the specular reflection of the surrounding water, supporting visual identification. This study found that the VV channel of the dual-polarized Sentinel-1 satellite offers the most visually and quantitatively distinct backscatter returns, as compared to the VH channel, when interacting with riverine mining dredges.

Two accuracy assessments were conducted: one, an absolute assessment of six points from WV02 imagery, and the other, a 75-point assessment from Planet Labs imagery. In comparing the two accuracy assessment rates, it appears that the combination of increased accuracy point collection and increased FP and FN results produced a lower total accuracy rate in the Planet Labs assessment. The single FP found in the WV02 assessment was caused by a large metal roofed building 30 m from the riverbank; strong backscatter from the building extended as far as the riverbank, leading to a false interpretation of a docked dredge. The most common reason for FP determinations in the Planet Labs assessment (five of 16 FP points for 24% of the total inaccuracy) was a similar misidentification of a building within 15 m of the riverbank. Riverbank interference, either concealing a dredge and leading to an FN or being misinterpreted as a dredge and leading to an FP, accounted for another four misidentifications in the data, amounting to 19% of total Planet-Labs-assessed inaccuracy. Four misinterpretations, for another 19% of the total inaccuracy, were caused by spoil piles returning a stronger signal than surrounding water values. A time difference of eight hours between the acquisition of the SAR data and the optical image, where it is possible the vessel had moved between acquisition times, accounted for another four misidentifications at 19% of the total inaccuracy. One misinterpretation (5% of the total inaccuracy) was caused by an instance of two or more vessels being close to each other and their signals blending together. There is no clear reason for the misidentification of the remaining three FPs (14% of the total inaccuracy), which presented as strong signals in the SAR data.

This study found that dredges and other riverine vessels are visible in 10 m SAR data with signatures distinctive enough to manually identify, confidently map, and temporally monitor for activity. However, the accuracy assessment shows that there are pitfalls to manual identification when unsupported by corresponding high-resolution optical imagery. Nevertheless, the number of FP designations outweighs the number of FN designations, indicating that the results of visual analysis are at least 72% accurate, erring on the side of inclusivity. There is thus a greater chance of overcounting dredges on the river than undercounting them; the method presented here leads to more inclusive coverage of potential hits rather than too-sparse results with potential for undercounting. The accuracy assessment also weighs the availability of optical data against SAR data and demonstrates that SAR is a reliable and consistent alternative to optical imagery, especially in remote or cloudy areas of interest.

*4.2. Identification Challenges*

Although consistent monitoring of riverine vessels was successful, the nature of SAR data ultimately limits the amount of information that can be recovered. One such limitation is the difficulty in differentiating between dredges and other vessel types, including barges, ferries, or tender vessels. Not every manual identification can be confirmed by high-resolution imagery. Another potential issue is signal conflation, such as when dredges operate in close proximity to each other, as seen in Figures 8 and 9. The double-bounce

backscatter from two or more dredges in close proximity blend together and may be interpreted as only one dredge. Similarly, dredges operating along riverbanks, as some SSM dredges tend to do, may be completely invisible in SAR data due to signal loss within the diffuse scattering created by riverbank features. The misidentification of man-made features on riverbanks as dredges can be addressed by referring to high-resolution imagery, but the inverse problem, wherein a dredge is not identified in a SAR scene and high-resolution imagery is therefore not checked at all, could lead to oversights and higher numbers of FNs. The utilization of different wavelengths may address these questions, where longer wavelengths may better penetrate riverbank overhang to provide a more comprehensive picture of the river surface.

Another challenge is the determination of vessel size. The largest dredges in this study area are 25–32 m in length. However, there is the possibility of smaller, perhaps artisanal dredges operating on rivers in other geographic contexts that may not appear in Sentinel-1 SAR data as a result of its spatial resolution. SAR data may yet have an advantage over optical imagery in such scenarios; double-bounce backscatter returns from dredges, even ones that are smaller than those seen in this study, can be so strong as to cause pixel over-saturation and bleed-over into neighboring pixels, increasing the footprint of the return to make the dredge appear larger than its true size, thus helping in the detection of smaller vessels. The growing availability of high spatial resolution SAR data, mainly through various new commercial SAR providers, could also help improve the detection of small vessels. Future studies can also benefit from increased SAR data availability and capabilities, which could present opportunities to differentiate between different types of dredges or vessels based on backscatter returns at high spatial resolutions, wavelengths, and/or polarizations.

Manually identifying riverine vessels in SAR data is time-intensive work that requires subject-matter expertise. Advanced machine learning techniques like convolutional neural networks and adaptive thresholding methods have shown success in detecting and classifying ocean ships [40,42,43,51]. These techniques may be transferable to the detection and monitoring of dredge mining activities to improve identification and decrease the time required by manual examination of each scene. Specifically, such techniques may be trained to efficiently identify salient spatial and temporal features from SAR data and then use them to categorize vessels and movement patterns.

### 4.3. Monitoring Riverine Dredge Activity

The results of the dredge detection and monitoring analysis show a year-on-year increase in activity, with the largest percentage increase occurring between 2016 and 2017. Similarly, both the average number and the maximum number of active dredges seen on the river in any given scene increased over time. The five-year period under review in this study began in 2015, when dredging was first introduced and the maximum number of dredges on the river at any point was two. The average number of dredges per scene was lower, at 1.1, indicating that there were no more than two active dredges operating on the river at any given time in 2015. The study concludes in 2019, immediately before the onset of a global pandemic that may have impacted results. That year, an average of 4.3 dredges were noted per scene, with a maximum of nine active dredges at any given time. Thus, the total number of dredges operating on the river likely went up by seven over five years. This suggests that river dredging increased significantly once initial dredging operations had been deemed successful and following the partial lifting of the KP export ban.

The largest concentration of mining occurred in the southern and central portions of the study area—regions that have a long record of terrestrial artisanal and small-scale diamond mining activity [5]. By 2019, dredges were present across the study area, with mining hotspots about 8 km north of Salo in the south/center and between Mayomba and Nola in the north. The intensity of mining activity is largely dependent on the geomorphology of the mined area. High activity frequently occurs between the type of geological ridgelines that can be seen in the southern half of the study area (Figure 14) [5]. Colluvial and fluvial

erosive processes erode the Carnot Sandstone of the region and transport diamonds to alluvial deposits [5]. The observed zones of high dredge concentration to the north of the study area are consistent with descriptions of sites that fluvially concentrate diamonds, gold, and coarse sediment in bedrock riffles due to slowing flows, making them valuable mining sites [5].

The results of the dredge analysis revealed areas of concentrated mining activity and shifts in activity over time. Insight into the location of riverine mine sites can help predict future dredge mining operations. Combined with geologic data, such dredge analyses may inform both current production estimates and the resource potential/production capacity of possible future mine sites. Dredge location data may also aid in the monitoring of ASM compliance with laws or restrictions on mining permissions. In addition, the environmental effects of dredge mining, which include sedimentation, in-channel spoil piles, and the possible use of mercury in gold amalgamation, could be better addressed by mapping and monitoring dredge activity [12]. These effects can spread far beyond the local mined area, influencing downstream pollution, sedimentation, and channel geomorphology, thus affecting flow and flood dynamics [4,18].

## 5. Conclusions

This study used Sentinel-1 SAR data to identify mining dredges on the Kadéï river in the subprefecture of Nola, Central African Republic, between 2015 and 2019. The analysis presented demonstrates the possibilities of implementing SAR in the monitoring of artisanal and small-scale riverine dredging activity. SAR has the advantage of large swath widths and high temporal frequency without the seasonal cloud-cover interruptions seen in traditional high-resolution optical imagery. The strong backscatter returns produced by metal vessels against the flat surface of a river create distinguishable visual cues for the identification of riverine vessels in SAR data. The validation of these dredges in corresponding high-resolution imagery increases confidence in the inclusivity of the results of the proposed method. Thus, manually identifying dredges in SAR imagery is possible and can provide reliable and repeatable results. However, this method requires time and skill in manual SAR interpretation. Further research into automated riverine vessel detection methods that adapt techniques used in ocean ship detection could advance both remote sensing techniques and applied research to improve the detection and monitoring of small-scale, sometimes informal mining activities in remote or conflict-prone terrain. The ability to identify and monitor riverine dredge mining hotspots is valuable both in the CAR and in remote or conflict-prone locations at large, where a dredge's elusive nature may confound conventional monitoring techniques.

**Author Contributions:** Conceptualization, P.G.C.; data production, K.L.O. and M.A.A.; writing—original draft preparation, P.G.C., M.A.A. and S.S.; writing—review and editing, S.S. and P.G.C.; visualization, M.A.A. All authors have read and agreed to the published version of the manuscript.

**Funding:** This research did not receive any specific grant from funding agencies in the public, commercial, or not-for-profit sectors.

**Data Availability Statement:** Feature locations of the detected dredges in this study are freely available as a point dataset on sciencebase.gov: Alessi, M.A., 2023, *Locations of small-scale diamond and gold dredges detected using Synthetic Aperture Radar on the Kadéï (Sangha) River, Central African Republic: U.S. Geological Survey data release*, https://doi.org/10.5066/P9FWFC7R (accessed on 17 November 2022).

**Acknowledgments:** The authors would like to recognize Kendall C. Wnuk (USGS) for valuable comments on an early version of the manuscript; Rob Stamm (USGS), Jessica D. DeWitt (USGS), and Kathleen M. Boston (Akima Systems Engineering) for review of data products and metadata; and three anonymous peer reviewers for their reviews and comments that helped to improve the manuscript.

**Conflicts of Interest:** The authors declare no conflict of interest.

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
