# Peer review of "Detection and Monitoring of Small-Scale Diamond and Gold Mining Dredges Using Synthetic Aperture Radar on the Kadéï (Sangha) River, Central African Republic"

_remotesensing, doi:10.3390/rs15040913_

Round 1
Reviewer 1 Report
This paper implements a remote sensing analysis of Synthetic Aperture Radar (SAR) data to map gold and diamond dredges operating on the Kadéï (Sangha) river in the CAR. The analysis presented demonstrates the possibilities of implementing SAR in the monitoring of artisanal and small-scale riverine dredging activity. In addition, two accuracy assessments were performed to measure the success of this paper’s methodology of manual detection of ASM dredges in SAR data. In general, this paper is well-written and can be published in its current form.
I have only one question as follows:
The Sentinel-1 SAR data is used in this paper. The differential InSAR technique can be used for deformation measurement. Can this technique be combined in this paper? If so, more information may be obtained.
Author Response
Reviewer 1: I have only one question as follows:
The Sentinel-1 SAR data is used in this paper. The differential InSAR technique can be used for deformation measurement. Can this technique be combined in this paper? If so, more information may be obtained.
Author response:
The differential InSAR technique could be helpful in combination with this type of study and should be looked into in the future. In this specific circumstance I believe the changes in the river would be too small to appear in the resolution of SAR that is available, however combining it to identify other mining going on around the river could open up a bigger picture of the overall mining in the area. No substantial change to make to the paper based on this comment.
Reviewer 2 Report
This work is an excellent technical report on the way in which the wealth (gold and diamonds) of CAR is abandoned in rivers of the country and in what way it can be collected.
The work does not contain any innovation or significant mathematical background. So from a scientific point of view it is of limited interest.
However, it provides for this country a complete study of the behavior of citizens as well as of foreign companies regarding the way of exploiting its mineral wealth. For this reason I recommend its publication in REMOTE SENSING in the present form.
This work is not only useful to the CAR government in case they want to use it for the benefit of the country, but it will also be useful to many neighboring countries.
The work is very well organized with completeness in the sections and the examples it presents. The references are adequate.
Author Response
Reviewer 2:
This work is an excellent technical report on the way in which the wealth (gold and diamonds) of CAR is abandoned in rivers of the country and in what way it can be collected.
The work does not contain any innovation or significant mathematical background. So from a scientific point of view it is of limited interest.
However, it provides for this country a complete study of the behavior of citizens as well as of foreign companies regarding the way of exploiting its mineral wealth. For this reason I recommend its publication in REMOTE SENSING in the present form.
This work is not only useful to the CAR government in case they want to use it for the benefit of the country, but it will also be useful to many neighboring countries.
The work is very well organized with completeness in the sections and the examples it presents. The references are adequate.
Author response: No substantial changes to address from this comment - we appreciate the assessment
Reviewer 3 Report
This study implements a remote sensing analysis of Synthetic Aperture Radar (SAR) data to map gold and diamond dredges operating on the CAR’s Kadéï (Sangha) river. Riverine vessels were identified in Sentinel-1 SAR data between 2015 and 2019, and their activity levels are mapped over time. The manuscript is clear, relevant to the field, presented well-structured, and scientifically sound. The manuscript’s results are reproducible based on the details given in the methods section. I think the paper needs in conclusion to mention more about their future work.
Author Response
Reviewer 3: This study implements a remote sensing analysis of Synthetic Aperture Radar (SAR) data to map gold and diamond dredges operating on the CAR’s Kadéï (Sangha) river. Riverine vessels were identified in Sentinel-1 SAR data between 2015 and 2019, and their activity levels are mapped over time. The manuscript is clear, relevant to the field, presented well-structured, and scientifically sound. The manuscript’s results are reproducible based on the details given in the methods section. I think the paper needs in conclusion to mention more about their future work.
Author response: We have mentioned future work in the conclusion section. We mention the need for further research into automated methods and how to further adapt ocean ship detection methods for small-scale mining in line 549-551. No substantial change to make